# The Simplicity Bias in Multi-Task RNNs: Shared Attractors, Reuse of Dynamics, and Geometric Representation

**Elia Turner**
Department of Mathematics
Technion, Israel Institute of Technology
eliaturner11@gmail.com

**Omri Barak**
Rappaport Faculty of Medicine and Network Biology Research Laboratory
Technion, Israel Institute of Technology
omri.barak@gmail.com

## Abstract

How does a single interconnected neural population perform multiple tasks, each with its own dynamical requirements? The relation between task requirements and neural dynamics in Recurrent Neural Networks (RNNs) has been investigated for single tasks. The forces shaping joint dynamics of multiple tasks, however, are largely unexplored. In this work, we first construct a systematic framework to study multiple tasks in RNNs, minimizing interference from input and output correlations with the hidden representation. This allows us to reveal how RNNs tend to share attractors and reuse dynamics, a tendency we define as the "simplicity bias". We find that RNNs develop attractors sequentially during training, preferentially reusing existing dynamics and opting for simple solutions when possible. This sequenced emergence and preferential reuse encapsulate the simplicity bias. Through concrete examples, we demonstrate that new attractors primarily emerge due to task demands or architectural constraints, illustrating a balance between simplicity bias and external factors. We examine the geometry of joint representations within a single attractor, by constructing a family of tasks from a set of functions. We show that the steepness of the associated functions controls their alignment within the attractor. This arrangement again highlights the simplicity bias, as points with similar input spacings undergo comparable transformations to reach the shared attractor. Our findings propose compelling applications. The geometry of shared attractors might allow us to infer the nature of unknown tasks. Furthermore, the simplicity bias implies that without specific incentives, modularity in RNNs may not spontaneously emerge, providing insights into the conditions required for network specialization.

## 1 Introduction

Consider the hand motions of a basketball player in a single game – dribbling, passing, receiving, and shooting. Each motion has its own dynamics, and yet all are orchestrated by the same area in the motor cortex. How can a single interconnected neural population give rise to a myriad of different actions? A recent body of work examined the link between task performance and neural dynamics by thinking of populations of neurons, especially in more cognitive and motor areas [5], as dynamical

37th Conference on Neural Information Processing Systems (NeurIPS 2023).

systems (see [31] for a full review). Trained Recurrent Neural Networks (RNNs) provide a natural model for cortical areas [1, 12, 27]. Anatomically, the cortex is enriched with lateral connections, making feedback the default architecture. Functionally, RNNs are universal approximators and can thus be trained on tasks similar to those studied experimentally. The downside of trained RNNs is that, as with any machine learning tool, they are often seen as a black box. Nevertheless, many advances have been made on understanding both the dynamics of trained networks [3, 4, 5, 12, 19, 21, 23, 29] and how these dynamics emerge through training [7, 26].

Most (but not all) of these investigations have been focused on one controlled task at a time. This focus, although valuable, may not fully capture the complexity of an animal's behavioral repertoire which is dynamic, diverse, and often involves multiple simultaneous tasks [14]. Shifting the focus from single to multiple tasks is a natural evolution in the field, aligning more closely with the realities of an animal's environment which is characterized by symmetries, regularities, and structures. It is highly plausible that the neural circuits driving behavior have adapted to this structure by developing efficient mechanisms to perform multiple tasks. This perspective has led to several studies investigating the behavior of neural networks in multi-task settings [16, 17, 33]. Experimentally, [8] demonstrated that different muscle activity patterns can be explained by neural activity modes in M1 that are task-independent.

Recent studies have begun to address the complex issue of multi-task performance using RNNs. [35] trained single network models to perform an array of 20 cognitive tasks that encompass working memory, decision making, categorization, and inhibitory control. They discovered that after training, recurrent units can develop into clusters that are functionally specialized for different cognitive processes. [6] further extended this line of research by identifying an algorithmic neural substrate for compositional computation. They conducted dynamical systems analyses of networks and found computational strategies that mirrored the modular subtask structure of the task-set used for training. Key dynamical motifs, such as attractors, decision boundaries, and rotations, were reused across different task computations. Their work contributes to the understanding of compositional computation, presenting dynamical motifs as a fundamental unit of computation that allows for flexibility and generalization of previously learned computations.

Our work offers a complementary approach to these pioneering studies. Instead of working from common tasks towards their implementation, we take a step back to consider the most fundamental dynamical objects typically found in RNNs. We ask from first principles: How can a single neural population combine these disparate objects into a joint representation? And under what conditions do these representations become shared or separate?

To explore these questions, we use RNNs as our computational tool and introduce a novel framework that disentangles cognitive computation from sensory and motor processes. By ensuring that all tasks operate under identical input and output statistics, our approach simplifies the concept of task-similarity, rendering it amenable for systematic analysis. Our results reveal a bias toward developing shared representations, which we term "simplicity bias". This tendency persists regardless of the number of tasks or the nature of the dynamical objects involved, be they fixed points, limit cycles, line- or plane- attractors. We link this bias to the sequential emergence of attractors during training, and show how external factors can resist this bias and create more complex dynamical objects. We continue to explore the inner structure of these shared attractors. By limiting the discussion to tasks that require line attractors, and re-conceptualizing tasks as functions, we show that the steepness of the computation, and not the computation itself, are responsible for shaping the attractor itself.

Our results can be used to infer hidden capabilities of networks by observing their activity geometries. The results also shed light on the conditions for the emergence of modularity in RNNs, and more generally provide a framework for understanding complex tasks in terms of dynamical building blocks.

## 2  The systematic study of multiple tasks

To systematically study multiple tasks, it is critical to ensure that our experimental design does not artificially influence the divergence or convergence of the network's representations. Our methodology is twofold:

First, we standardize the input and output structure across all tasks and use a consistent network architecture. This approach ensures that any observed differences in representations can be attributed to intrinsic task complexities, rather than artificial variations in input-output structure or network design. This methodology is inspired by the discussion in [34], which highlights that single-layer RNNs, unlike the brain, simultaneously implement sensory, cognitive, and motor elements. This all-in-one implementation makes it challenging to attribute different parts of the computation to distinct elements. By standardizing the input-output structure and network design, we aim to sidestep this issue and gain a clearer understanding of the nature of task representations.

Secondly, to guard against artificial convergence in representations, we assign individual input and output channels for each task. This design choice ensures that any observed overlap or similarity in representations results from shared computational requirements or reused dynamics across tasks, rather than imposed sharing of input or output channels. Following these principles, we outline our experimental setup as detailed below.

## 2.1 Model description

We use a discrete-time recurrent neural network (RNN), with $N$ hidden units. The state at each time step $t$ is given by:

$$h_{t+1} = \tanh(W_{rec}h_t + W_{in}u_t + b_{rec}) \tag{1}$$

where $W_{rec}$ is the recurrent weight matrix, $h_t$ is the hidden state at time t, $W_{in}$ is the input weight matrix, and $u_t$ is the input at time t, and $b_{rec}$ is a bias term. The network output, $y_t$, is:

$$y_t = W_{out}h_t + b_{out} \tag{2}$$

where $W_{out}$ is the output weight matrix and $b_{out}$ is the bias term. For full details about the training process see supplementary 2.

## 2.2 Task structure

For more details see supplementary section 1.

We consider a set of tasks designed to elicit certain dynamical objects in the trained network [28]. The input sequence for each task consists of pulses with variable amplitude, interspersed with periods of no input. Tasks are defined based on the output during the intervals between inputs (Fig. 1A). For instance, requiring the network to maintain one of two discrete values (Fig. 1A, top) results in the network producing two fixed point (Fig. 1B, top). In a similar manner, different input-output demands result in limit cycles, line attractors, or plane attractors (second, third and fourth rows respectively).

Formally, the network receives a sequence of input values $(a_1, ...a_{n-1}, a_n, ...)$ at corresponding times $(t_1, ...t_{n-1}, t_n, ...)$ such that:

$$\forall i \geq 0 : a_i \in [v_{min}, v_{max}], \quad t_{i+1} - t_i \in [d_{min}, d_{max}] \tag{3}$$

The output at time $t$ maps the recent history $(a_n, a_{n-1}, ..., t_n, t_{n-1}, .., t)$ onto the interval $[v_{min}, v_{max}]$. The specifics of this mapping influence the type and size of working memory required by the task, with any dependence on $t$ determining the temporal structure of the output. In the neuroscience literature, line attractors are commonly associated with analog working memory [11, 20], fixed points with decision making [32], and limit cycles with motor patterns [22]. In general, however, there is more than one way to implement a given task [30].

## 2.3 Architecture and training procedure

For full details see supplementary section 2.

Our main objective in multi-task learning is to investigate the internal representations formed by the RNN. The choice of architecture can have a dramatic effect on these representations. For instance, if two tasks share a common output, then the internal state of the network has to be different for different tasks by construction. In contrast, if each task has an independent output, then the internal representation may utilize similar states and it may utilize different states. We focus on an architecture that maintains this freedom, but note that recognizing and characterizing these architectural biases is also an important research direction. Our setup includes separate input and output channels for each

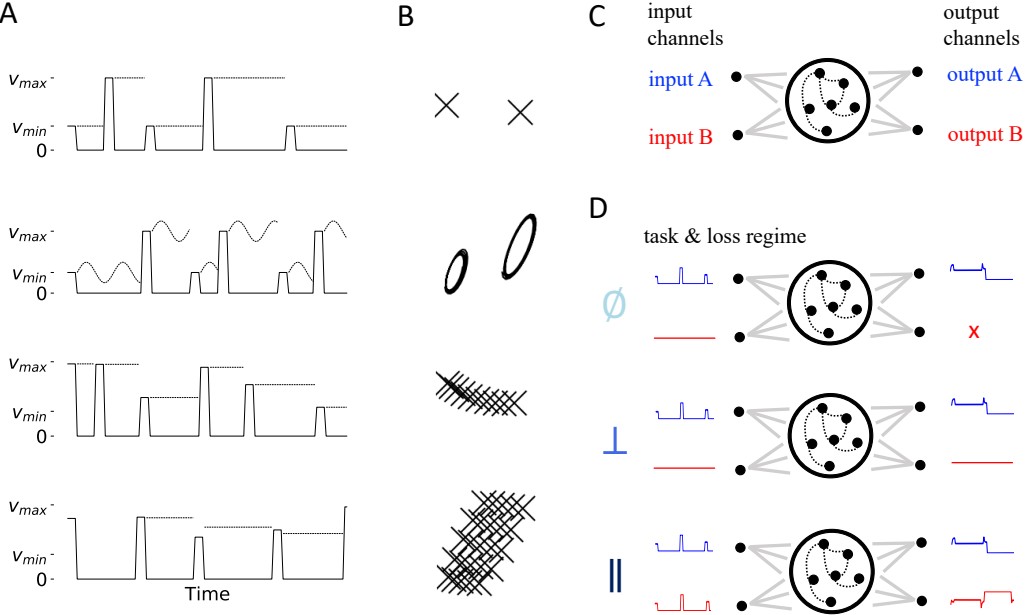

Figure 1: Multi-task learning of dynamical objects. **A** Tasks used in the paper consist of a series of pulses of varying amplitudes (solid lines). The targets are denoted by dashed lines. **B** Dynamical objects created by the various tasks. Examples of trajectories (or fixed points) in the first two principal components. From the top: two fixed points, two limit cycles, a line attractor, and a plane attractor. **C** Network architecture. A recurrent network receives two inputs and produces two outputs. **D** Training regimes. In the gated ($\emptyset$) and orthogonal ($\perp$) settings only one task is active at a time, with the other output being irrelevant (gated) or zero (orthogonal). In the parallel ($\|$) setting, every trial combines both tasks in random times.

task, so that $W_{in} \in \mathbb{R}^{N \times N_{tasks}}$ and $W_{out} \in \mathbb{R}^{N_{tasks} \times N}$ (Fig. 1C). We will denote by $N$ the number of hidden units.

Apart from the network architecture, the trial structure of multi-task training also has several degrees of freedom. We consider three regimes: gated, orthogonal, and parallel (Fig. 1D, marked respectively with $\emptyset$, $\perp$, $\|$). In the gated scenario, every trial has inputs from only one task, and the loss is computed for the output of that task alone, ignoring (or gating) the other task outputs. For the orthogonal case, inputs are still separated in each trial but the output of the irrelevant tasks has to be zero. Finally, in the parallel setting, both tasks are presented simultaneously within the same trial.

## 3 RNNs representations show a simplicity bias

We trained 30 networks on pairs of tasks in the gated and orthogonal settings (Fig. 2, parallel settings were tested for some of the combinations, see supplementary 5) . For visualization purposes, we projected the states $h_t$ onto their leading two principal components. We found that in the gated setting, the attractors of the two tasks shared similar representations, whereas in the orthogonal case the attractors were much more separated. When the two tasks demanded similar objects (columns 1, 4, 6 and 7) the gated setting resulted in a complete overlap between the tasks. Even when the attractors differed qualitatively, we still observed that they reside in similar areas of phase space. To quantify this effect, we trained a linear classifier to separate the neural trajectories of one task from those of the other one. For the gated setting, classification failed in nearly all networks (Fig. 2B, see supplementary 3.3), suggesting that the networks generated a single attractor with parts of it serving each task. This was also evident in the ratio of variances between and within tasks (F-factor, Fig. 2C, see supplementary 3.4). In contrast, the orthogonal setting resulted in separate attractors for each task. Overall, we conclude that network dynamics evolve according to a "simplicity bias", where the simplest dynamical structure that is sufficient for the joint solution of all tasks is the one that will emerge. This bias is present even for attractors that might seem incongruent, such as limit cycles and

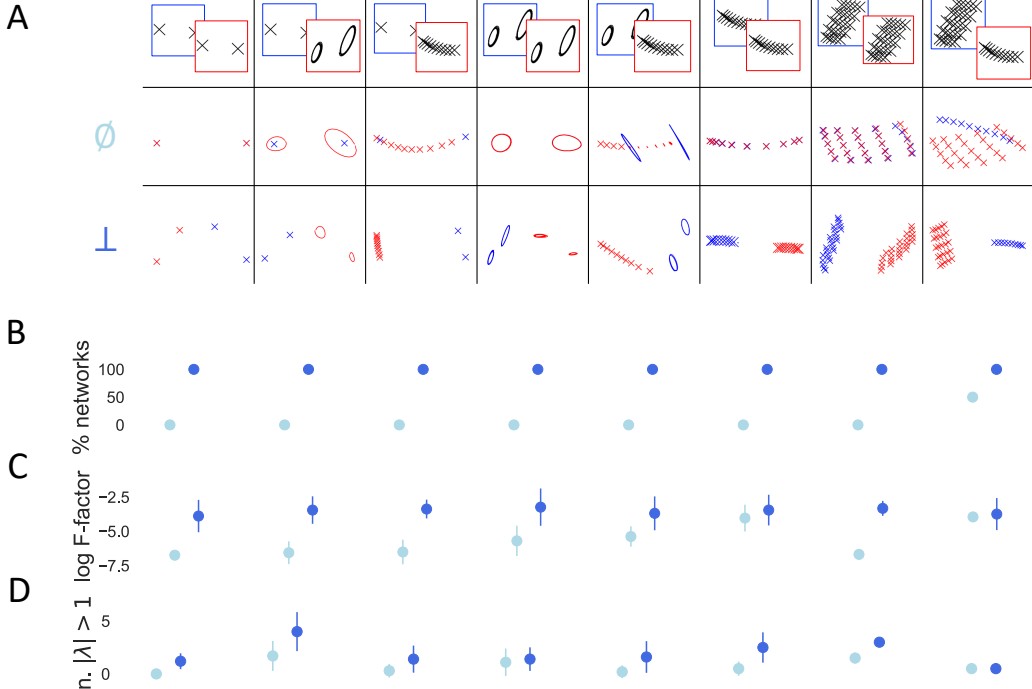

Figure 2: Simplicity bias. **A** Networks were trained on pairs of tasks in the gated or orthogonal settings. Neural activity in the first two principal components is shown for an example network of each combination. Note that gated (∅) networks reuse the same attractors for both tasks. **B** Fraction of networks for which the attractors of the two tasks could be linearly separated in phase space. **C** The F-factor is the ratio of the variance between attractors and within attractors. **D** Number of unstable modes of the connectivity matrix.

fixed points. To demonstrate that a bias, and not a hard constraint, we used the orthogonal solutions as initial conditions for training in the gated setting. Retraining in the gated setting with added noise did not shift the solution from that of the orthogonal one.

Because dynamics stem from the recurrent connectivity, we hypothesized that the elaboration of the dynamics in the orthogonal setting is reflected in the spectrum of the connectivity matrix. Indeed, we found that the number of unstable eigenvalues was larger in the orthogonal setting for all tasks (Fig. 2D, see supplementary 3.5).

The above results lead to a simple hypothesis regarding the origin of the simplicity bias (Fig. 3). Consider a pair of tasks both requiring fixed points. Initially, only the origin is stable, and the inputs from both tasks lead to states that converge to the origin. The resulting loss modifies the recurrent connectivity and dynamics. Eventually, a stable fixed point will emerge away from the origin. Once this happens, the inputs from both tasks will lead to states that converge to this new fixed point. Crucially, the recurrent dynamics cause the states to mostly depend on the task-agnostic attractors, and less on the task-specific inputs. If the loss to both tasks can be reduced within this attractor, then learning is not expected to generate another one. Importantly, even if another attractor is needed (as in the fixed point and limit cycle combination), we expect it to emerge in the vicinity of the existing attractor. This is because gradient descent is proportional to the state of the network, modifications to the connectivity will be proportional to this state . In the orthogonal setting, the two attractors have to be orthogonal to each other, forcing the network to separate them.

We verify this hypothesis by following the attractor landscape of networks as they train. To better characterize the dynamics, we use a low-rank network [13]: $W_{rec} = mn^T$, where $m, n \in \mathbb{R}^{N \times 2}$ are the only parameters trained via gradient descent, constraining connectivity to be of rank 2. A consequence of this connectivity is that dynamics can be visualized in a two-dimensional space given by the projections $\kappa_i(t) = n^T h_t$ (the definition is slightly different from that of [13] because of the

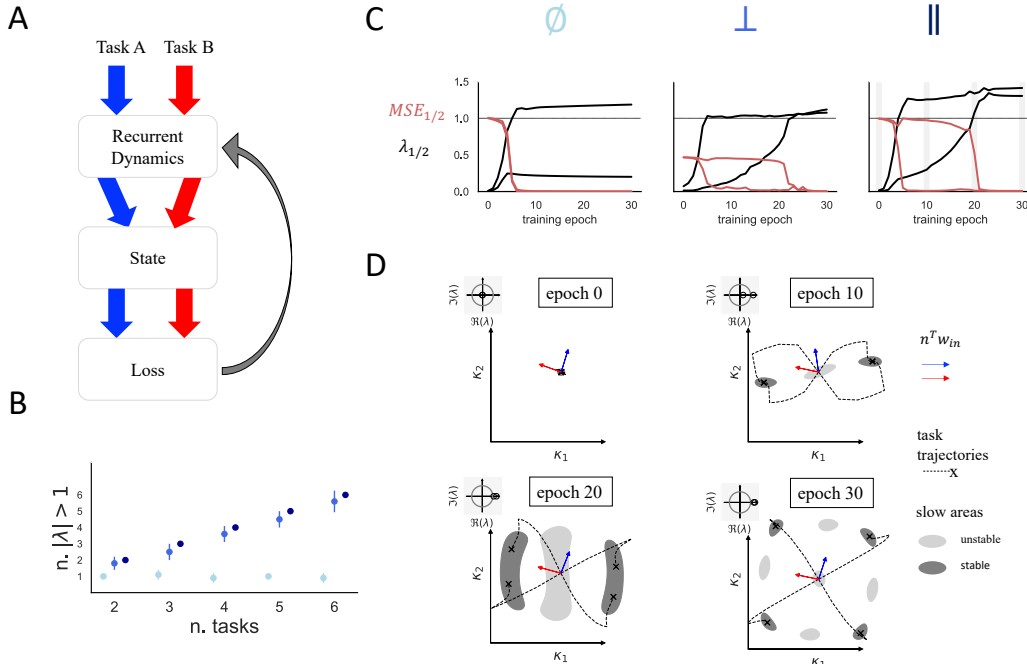

Figure 3: Sequential emergence of attractors underpins simplicity bias. **A** Cartoon illustrating the hypothesis. Inputs from both tasks undergo the same recurrent dynamics. Once an attractor forms from one task, the second task's activity gravitates towards it. New dynamical objects only form when task conditions require another attractor. **B** The number of unstable directions of the dynamics at the origin (eigenvalues of $W_{rec}$ with norm larger than 1) as a function of the number of tasks (bits) trained on. Gated networks, with their shared attractors, only develop a single unstable eigenvector. Orthogonal and parallel networks show a correlation between the number of attractors and unstable eigenvalues. **C** Reduction of error (red) and evolution of eigenvalues (black) over the course of training. Three rank-two networks were trained on a two-bit fixed point task using different setups. Note that the drops in error are accompanied by an eigenvalue obtaining a norm larger than one (becoming unstable). **D** Snapshots from the training process for two fixed-point tasks in the parallel settings (as in C-right, snapshots correspond to gray stripes there). Each snapshot shows the state space in two dimensions ($\kappa$). Gray shadings denote slow areas of the dynamics. Colored arrows denote input directions. Insets show the two eigenvalues, with the unit circle marking the stability border.

differences in the definitions of the dynamics). See also supplementary 3.6. We follow the evolution of the dynamics for such a network trained on two fixed-point tasks in the parallel setting (Fig. 3D). Initially, only the origin is stable, as reflected in the two eigenvalues of $W_{rec}$ being inside the unit circle. Eventually, the origin destabilizes, and a single unstable eigenvalue emerges along with a pair of stable fixed points (due to symmetry of the dynamics). With more training, the second eigenvalue leaves the unit circle, and shortly after another pair of stable fixed points emerge, leading to the final attractor landscape with four basins of attraction.

We repeated this analysis for a two-task version (all fixed points) in all three settings (Fig. 3B). We show the loss for each task and the spectrum of eigenvalues. In the parallel and orthogonal settings, one can see the clear sequential emergence of two outliers, accompanied by decreases in loss for each sub-task. The gated setting is solved with a single outlier, as all tasks share a common attractor.

A systematic analysis of many such tasks (sets of fixed points, from 2 to 6 simultaneously) shows that the number of outliers is proportional to the number of tasks for the orthogonal and parallel, but not for the gated setting. We repeated this analysis for both GRU networks and Vanilla networks initialized with a rich output regime and obtained similar results. See supplementary 3.7 for details.

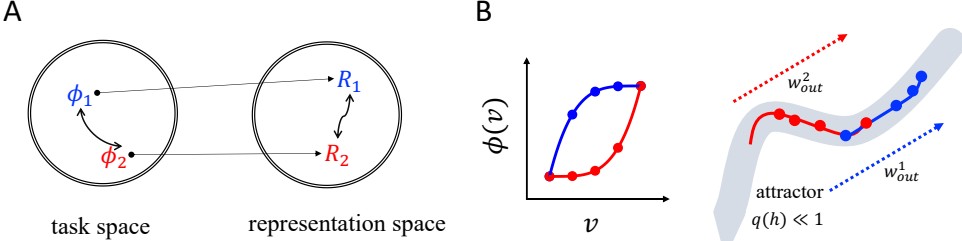

Figure 4: Task-Representation Relationship Analysis. **A** An abstract depiction of the task space (left) and representation space (right). Within the task space, two tasks (colored in red and blue) are illustrated, while their corresponding representations are delineated in the representation space. **B** Tasks are conceptualized as functions, and their representation is envisaged as a location on a joint line attractor. On the left, two 1D functions are valued on a uniform grid. On the right, the possible joint representation of these functions is shown along the joint attractor, with dots indicating the locations of the input grid.

## 4 Task similarity to representational similarity

Having established how RNNs tend to implement a shared attractor for all tasks, we aim to explore the principles that govern how multiple tasks will occupy the same attractor. In particular, how distances in task space translate to distances in representation space (Fig. 4A). To test this, we focus on a family of tasks that all require a line attractor.

### 4.1 Tasks as Functions

In line with our task structure (Section 2.2), the output of each task is a mapping $\phi : [v_{min}, v_{max}] \rightarrow [v_{min}, v_{max}]$, which operates on the most recently seen input $a_n$. This form enables us to conceptualize a set of tasks as a family of 1D parametric functions $\phi_i$. An example for two such functions can be seen in 4B (left). Transitioning to a function space allows us to define measurable quantities mathematically and observe how they translate into representation.

### 4.2 Neural representation of tasks

To probe the relationship between task similarity and representation similarity, we trained rank-2 RNNs on six such tasks:

$$\phi(v) = (1-v)^4, (1-v)^2, 1-v, v, v^2, v^4$$

and generated the input similarly to what was explained in Section 2.2. Utilizing the rank-2 constraint, we are able to visualize the joint representation of all six tasks for two network realizations. The contour lines of Fig. 5A denote the velocity of the RNN dynamics ($q = |h_{t+1} - h_t|$, indicating the position of the line attractor. In both realizations, all of the tasks (colored dots) occupy the same attractor, consistent with the results of the previous section (illustrated in Fig. 4B).

### 4.3 Arc length parameterization of the attractor

To quantify the similarity of representations, we parameterize positions along the line attractor. For any given task $i$, we define its representation $R_i$, as the ensemble of all achievable steady states. For each value $v$, uniformly spaced between $v_{min}$ and $v_{max}$, we examine the corresponding steady state, denoted by $R_i(v)$ that belongs to $\mathbb{R}^N$. This results in a curve $\{R_i(v)\}_v$, a one-dimensional attractor embedded within an N-dimensional space. Because these are rank-2 networks, it is actually embedded in a 2-dimensional space, but we don't rely on it for this analysis.

Consider the entirety of points across all attractors $\{r_j\}_i$, which are sorted w.r.t. their distance to a given starting point $r_0$. This arc length is determined as:

$$s(r_0) = 0, \quad s(r_{j+1}) = s(r_j) + \|r_{j+1} - r_j\|_2$$

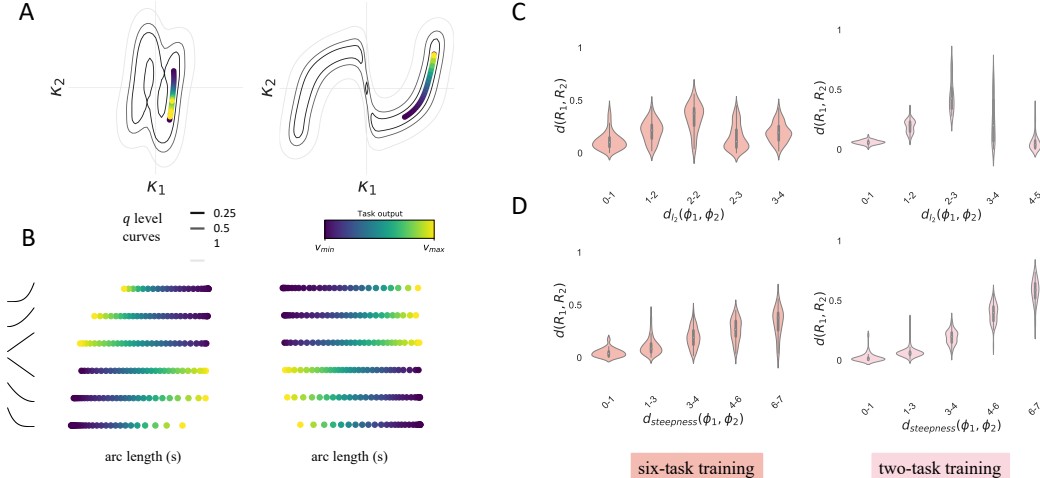

Figure 5: The joint representation of multiple tasks as a line attractor.**A.** Two exemplar *kappa*-projections of all attractors, with differing grey shades (from dark to light) representing different level curves of the $q$-function ([28], see supplementary 2.2). **B** Attractor projections into a 1D space using arc-length parameterization, stacked upon one another and aligned to the generating functions on the left. The points are colored w.r.t. the task output. **C** pairwise distance between each pair of task representations in every network, binned w.r.t. $l_2$ distance. On the left we show the results for networks trained on six tasks, and on the right for networks that were trained only on two tasks at a time. **D** same but binned w.r.t. the steepness distance.

After this computation, the resultant arc lengths are normalized to fit within the $[0, 1]$ range. This normalization produces the arc length function $s(R_i(v))$ which signifies the position of $R_i(v)$ on the unified one-dimensional curve.

In Fig. 5B we stack the 1D projection of the curves of each task, where on the left the corresponding functions appear. One can clearly see a gradual change in the functions, paralleling a gradual change in their representation.

## 4.4 Task distance

The functions in Fig. 5B seem to be ordered, but it is not immediately clear what is the metric underlying this ordering. A priori, there isn't one *correct* way to define the distance between tasks. When working with functions, the default way is to consider the $l_2$ distance between the corresponding functions:

$$d_{l_2}(\phi_1, \phi_2) = \int_{v_{min}}^{v_{max}} \|\phi_1(v) - \phi_2(v)\|_2^2 dv \qquad (4)$$

Observing the ordering emerging from the networks, an alternative way is measuring the similarity between the steepness of the functions:

$$d_{steepness}(\phi_1, \phi_2) = \int_{v_{min}}^{v_{max}} \||\phi_1'(v)| - |\phi_2'(v)|\|_2^2 dv \qquad (5)$$

In practice, we approximate these integrals using Reimanns sum formula.

## 4.5 Representational distance

Representational distance quantifies the overlap between two distinct task representations. Leveraging the arc length function $s$, the distance is defined as:

$$d(R_1, R_2) = 1 - \frac{|\{s(R_1(v)\}_v \cap \{s(R_2(v)\}_v|}{|\{s(R_1(v)\}_v \cup \{s(R_2(v)\}_v|} \qquad (6)$$

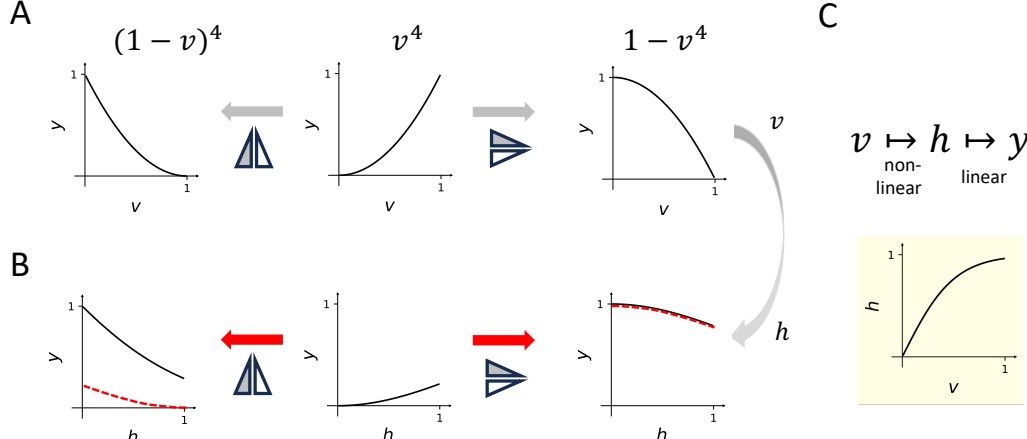

Figure 6: Intuition for symmetries in attractor representations, using a simplified 1D depiction. **A** A task defined by the function $y = v^4$ (middle) and two functions obtained by horizontal and vertical reflections. **B** The same three functions, now depicted as a function of $h$. The red lines are reflections of the transformed $v^4$, where it is evident that only the vertical reflection is conserved. **C** The nonlinear transformation from $v$ to $h$ (which includes the dynamics of the network) is the reason that horizontal reflections of a task yield different representation.

### 4.6 Function steepness shapes representations

To asses which task similarity correlates more with the representation similarity, we compute all pairwise distances between tasks - both in task space, and in representation space. The left columns of Fig. 5C,D show the relationship between either function-distance (C) or derivative distance (D) and the representation distance. The derivative distance is much more predictive, although not perfect. The existence of six simultaneous tasks could explain some of the spread in these graphs, so we also analyzed a simpler scenario. We trained full-rank networks on one pair of tasks at a time (right columns of Figure Fig. 5C,D). In this case the correlation between derivative distance is much stronger.

### 4.7 Symmetries in attractor representations

Why is it that steepness controls task alignment within attractors? A different perspective is to consider symmetries and invariances in task- and in representation- space [9]. Consider one specific function, $y = v^4$, depicted in the center of Fig. 6A. One can apply two different symmetry operations on it to obtain different functions. We know from the results of Fig. 5C,D that only the *vertical* reflection will preserve steepness, and hence is expected to lead to a similar representation. To gain intuition on why this happens, consider the transformations from input to output (Fig. 6C). Input undergoes a nonlinear transformation to hidden state – both due to static nonlinearities and due to the dynamics leading to the corresponding fixed point. From the hidden state to the output, however, is a linear transformation. We ignore the different input and output directions, and the curvature of the attractor and encapsulate the entire transformation as some nonlinear function – for instance, the one in Fig. 6C. If we now replot the three functions in terms of their relation to the hidden state (Fig. 6B), we see why only transformations that respect steepness lead to similar representations.

## 5 Discussion

We showed that recurrent neural networks trained on multiple tasks exhibit a simplicity bias in terms of the dynamical objects formed during training. Unless prevented, networks will form the minimal number of objects needed to solve the task. We demonstrated that this bias is a consequence of recurrent dynamics and the sequential formation of attractors. First, attractors are generated one at a

time. Second, once an attractor forms, the recurrent dynamics will converge to this attractor for all tasks.

Our analysis focused on an architecture with separate input and output channels for each task, and without an explicit context channel. Furthermore, the input and output statistics were identical for all tasks. Several studies of multi-task learning used different architectures. [10] showed how modular representations can emerge from multi-task learning in feed forward networks. Their single input - many output configuration results in a joint representation of all tasks, with the nature of this representation being the object of study. Specifically, they showed that random classification tasks disentangle the representation. RNNs trained on many cognitive tasks with more complex input and output structures also show shared attractors [6, 35]. The task-set analyzed there is closer to neuroscience experiments, making it easier to compare to data. On the other hand, these tasks have a correlation between their input-output structure and their cognitive demands. Here, we opted for simpler tasks, using fundamental dynamical objects, and with identical input-output structure for all tasks. This allowed a systematic study of how dynamical objects arise during training and relate to one another.

Our choice for simple and systematic tasks also comes with several limitations. The tasks are not as close to biology as those in previous works [2, 35]. On the mahine learning side, the tasks and architectures are very far from state of the art. While we did examine GRU architectures and lazy vs rich initializations, the focus here was on the systematicity and to remain close to elementary dynamical objects.

The underlying mechanism for the simplicity bias is the sequential emergence of attractors, and the reduction of loss once enough attractors have emerged. This is reminiscent of several results showing the emergence of various network properties at different rates. Connectivity in deep linear networks is affected by leading modes of the data covariance faster than other modes [25]. RNNs trained on fixed point tasks develop rank-1 connectivity faster than higher ranks [26]. Gradient descent first learns low moments, and then high cumulants of the data [18]. In modular feed forward networks, different pathways through the network *race* with each other, leading to a reduced effective network [24].

The simplicity bias has several intriguing applications. First, reusing an existing representation leads to faster learning of tasks. This feature was most evident in the gated scenario, in which only one task is processed at a time. [15] highlighted a similar phenomenon as a possible normative reason for the limited capacity of cognitive control. Second, our results suggest that observing a modular representation is an indicator of some constraint on the architecture. Finally, an attractor shared between tasks is not identical to an attractor that was formed in response to a single task. For instance, if a task with constant output has an oscillatory component to its neural activity, it might suggest that the same circuit is also capable of generating such an oscillation in another context. One can imagine recording the neural activity of a subject dribbling a basketball, and being able to predict whether that subject also knows how to shoot.

## Acknowledgments and Disclosure of Funding

This work was supported in part by the Israeli Science Foundation grant 1442/21 and an HFSP research grant (RGP0017/2021).

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
