A python code for the figures and results is partially available at `https://github.com/eliaturner/multitaskrepresentation`.

# 1 Task and trial structure

## 1.1 Individual tasks

The design of tasks is elaborated in Section 2.2 of the main paper. Task inputs consist of a sequence of pulses, each pulse being present for 5 steps with $a_n$ as the pulse's magnitude. Subsequent to every input pulse, a delay interval selected from the range $[d_{min}, d_{max}]$ is employed before the next pulse emerges. Across all experiments, we fix $d_{min} = 10$ and $d_{max} = 40$. The table presented below delineates all pertinent details regarding the construction of individual tasks:

| Attractor type | Figures | Input range | Structure | Trial duration |
|---|---|---|---|---|
| 2 fixed points | $1B, 2$ | $\{1, 3\}$ | $y(a_n) = a_n$ | 200 |
| 2 limit cycles | $1B, 2$ | $\{1, 3\}$ | $y(a_n, t) = 0.5 \cdot \sin\left(\frac{4\pi t}{d_{max}}\right) + a_n$ | 200 |
| line attractor | $1B, 2$ | $[1, 3]$ | $y(a_n) = a_n$ | 200 |
| plane attractor | $1B, 2$ | $[1, 3]$ | $y(a_n, a_{n-1}) = a_n + a_{n-1}$ | 200 |
| 2 $n_{bits}$ fixed points | $3$ | $\{-1, 1\}$ | $y(a_n) = a_n$ | 200 |
| line attractor | $5$ | $[1, 3]$ | $y(a_n) = 2 \cdot \phi\left(\frac{a_n - 1}{2}\right) + 1$ | 250 |
| | | | $\phi(v) = v^k, \quad k = 1, 2, 4$ | |
| | | | $\phi(v) = 1 - v^k, \quad k = 1, 2, 4$ | |
| | | | $\phi(v) = (1 - v)^k, \quad k = 2, 4$ | |
| | | | $\phi(v) = 1 - (1 - v)^k, \quad k = 2, 4$ | |

## 1.2 Combining multiple tasks

Thus far, the input and output structure of individual tasks has been discussed. Following, we examine how single tasks can be integrated into a multiple task setup. In this paper, we only explore scenarios where each task possesses its own distinct input and output channel. Specifically, the dimensionality of both the input and output training tensors was $[n_{trials}, n_{steps}, n_{tasks}]$.

Given every task has its own output channel, there are at least three setups to consider. Firstly, a *gated* setup, denoted by $\emptyset$, where each trial focuses on a single task, while the output of other tasks is disregarded. Secondly, the *orthogonal* setup, symbolized by $\perp$, also concentrates on one task per trial, but mandates the remaining outputs to be 0. Finally, the parallel ($\parallel$) setup, where all tasks are active simultaneously in every trial.

Essentially, the $\emptyset$ trials are encompassed in both $\perp$ and $\parallel$ setups. Trials of $\perp$ can be regarded as being contained in the $\parallel$ setup, particularly for trials where only one channel is active.

# 2 RNNs architecture and training

## 2.1 Network Architecture

The Vanilla RNN [1] with $N = 100$ units is utilized throughout the paper.

$$h_t = \tanh\left(W_{ih}ut + b_h + W_{rec}h_{t-1}\right) \tag{1}$$

The readout is given by:

$$\hat{y}t = W_{out}h_t + b_{out} \tag{2}$$

For some segments, the matrix $W_{rec}$ was of rank-2, a product of two matrices $W_{rec} = mn^T$, where $m, n \in \mathbb{R}^{N \times 2}$.

To train the rank-2 networks [2] in Figures 4 and 5, full-rank RNNs were initially trained on the given task, and then the final weights and the rank-2 approximation of the resulting $W_{rec}$, denoted as $W_{rec}^2$, were used:

$$USV^T = W_{rec} \rightarrow W_{rec}^2 = s_{1,1}u_{:,1}v_{1,:} + s_{2,2}u_{:,2}v_{2,:}$$

These served as initial conditions for training a rank-2 model on the same tasks. This model is denoted as Vanilla→Rank2.

## 2.2 Trainable Parameters and Weight Initialization

The parameters $W_{out}, b_{out}$, and $b_{rec}$, when used, were initialized from the uniform distribution $U\left(-k, k\right)$, where $k = \frac{1}{\sqrt{N}}$. For the full rank version, $W_{ih}$ was also initialized from $U\left(-k, k\right)$, and $W_{rec}$ was initialized from $U\left(-k \cdot g, k \cdot g\right)$, with $g = 0.5, 1$. For the rank-2 version, $W_{ih}$ was initialized from $U\left(-k, k\right)$, and $m$ and $n$ were initialized from $\mathcal{N}\left(0, k\right)$. Throughout the paper, the parameters $W_{rec}, b_h, W_{out}$, and $b_{out}$ were trained. In some instances, we also trained $W_{in}$. In certain cases, the recurrent bias term was excluded. The subsequent table specifies the model, parameters used, and initialization used for each figure.

|  | Model type | Train $W_{ih}$ | $b_h$ | No. of nets | $g$ | Task regime |
|---|---|---|---|---|---|---|
| $1B$ | Vanilla | $X$ | $V$ |  | 0.5 | $\emptyset$ |
| $2A, B$ | Vanilla | $X$ | $V$ | 30 |  | $\emptyset, \perp$ |
|  |  |  |  | 10 | 0.5 |  |
|  |  |  |  | 20 | 1 |  |
| $2C, D$ | Vanilla | $X$ | $V$ | 10 | 0.5 | $\emptyset, \perp$ |
| $3B$ | Vanilla | $X$ | $X$ | 10 | 0.5 | $\emptyset, \perp, \parallel$ |
| $3C, D$ | Rank-2 | $X$ | $X$ |  |  | $\parallel$ |
| $5$ | Vanilla → Rank-2 | $V$ | $X$ | 25 | 1 | $\emptyset$ |
| $5C, D$ | Vanilla | $V$ | $X$ | $8 \times 45$ | 1 | $\emptyset$ |

## 2.3 Training protocol

All networks were trained using the *Adam* optimizer [3] for $10,000$ epochs with a batch size of 32. The learning rate was initially set to $1 \times 10^{-3}$ and decayed

until it reached $1 \times 10^{-5}$. Unless specified otherwise, the training set consisted of 400 trials, and the order of these trials was shuffled at the start of each epoch. The performance of the network was evaluated using mean squared error (MSE), and the training was stopped once a minimum threshold of $10^{-4}$ was achieved over the training set.

### 2.3.1  Task and trial structure

For all tasks, the network was trained only on steps that did not include an external input.

## 3  Data analysis

### 3.1  Plotting alternative setups

In Figure 1, we provide illustrative examples demonstrating how each of the task setups, the *gated*, *orthogonal*, and *parallel*, lead to qualitatively different joint representations:

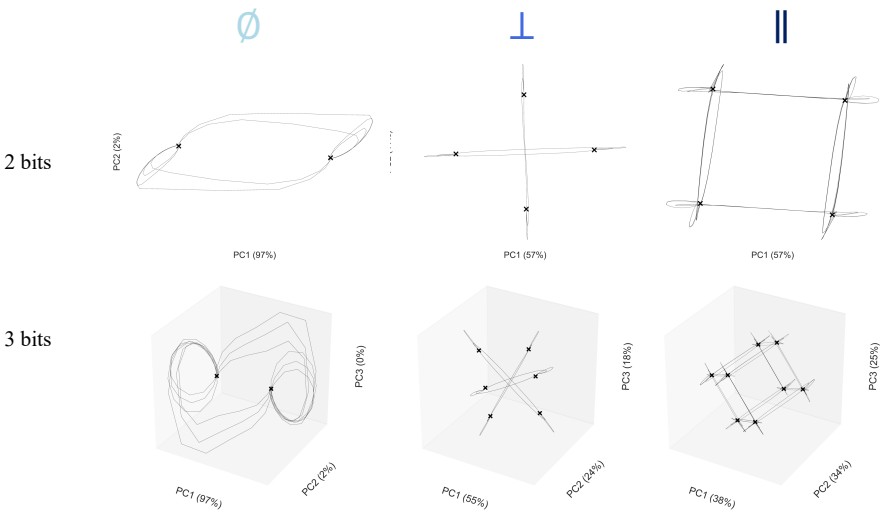

Figure 1: The neural representation of the flip-flop task for the *gated* ($\emptyset$), *orthogonal* ($\perp$), and parallel ($\parallel$), for either two or three bits. For the two-bit scenario, we projected into the first two principal components (PCs), and for the three-bit scenario, we projected into the first three PCs. Each plot shows the percentage of the variance explained by each dimension.

During training, each input value is generated from a uniform distribution. However, during validation trials, we only generate values from a predefined sequence of values:

$$v \in \mathcal{V} := \left(v_{\min} + \frac{l-1}{n_{\text{values}} - 1} v_{\max}\right)_{l=1}^{n_{\text{values}}}$$

(3)

where $n_{\text{values}}$ is the number of desired samples. In our reports, we set $n_{\text{values}} = 15$.

## 3.2  Attractor extraction

For each task, we ran the dynamics for inputs sampled from the validation set presented above. For every value, we allowed the dynamics to evolve autonomously and collected the states reached when the neural speed was less than $10^{-4}$. We quantified the neural speed in phase space [4] as

$$q(h_t) = \|h_{t+1} - h_t\|_2, \quad t \in \mathbb{N}$$

For every condition, such as different input history, we averaged over the set of states. When the network reached a limit cycle, we associated each input condition with a complete cycle. At the end of the day, for every task, we had a list of attractor states that span all possible input conditions.

## 3.3  Measuring separability

In Figure 1, for every pair of tasks, we trained a linear SVM to distinguish between the attractor states of the tasks. In the *gated* setup, the tasks were inseparable in all instances. In the *orthogonal* setup, all of the tasks were separable. We normalized all of the states jointly before inputting them into the algorithm. We used the *sklearn.svm* method with parameter $C = 100$.

## 3.4  F-factor

To assess the differences between the representations of two tasks, we utilized a statistical method that compares the variances within and between the sets. This approach, often used in statistical analysis, calculates the variances within each set separately, $\text{var}(R_1)$ and $\text{var}(R_2)$, as well as the variance of the combined set $\text{var}(R_1 \bigcup R_2)$. From these measurements, we define the within-group variance as the mean of the two variances:

$$V_{within} = \frac{\text{var}(R_1) + \text{var}(R_2)}{2}$$

The variance between groups is then defined as the difference between the combined variance and the mean within-group variance:

$$V_{between} = \text{var}(R_1 \bigcup R_2) - V_{within}$$

We then calculate the F-factor, denoted as $F$, by dividing the between-group variance by the within-group variance:

$$F = \frac{V_{between}}{V_{within}}$$

The F-ratio serves as a measure to assess the significance of the observed differences. A larger F-ratio indicates a higher likelihood of significant differences between the sets, while a smaller F-ratio suggests that the variation within the sets is relatively higher compared to the variation between them. This analysis is shown in Figure 2.C.

## 3.5 Spectrum analysis

For each network, we computed the spectrum of the eigenvalues of the recurrent matrix $W_{rec}$. In Figure 1, we counted the number of eigenvalues with norm greater than 1 for each network. In discrete systems, eigenvalues with a norm greater than 1 are unstable. The spectrum, however, is not a complete description for a nonlinear system. It does describe the dynamics around the origin (in the absence of bias), and can therefore indicate when unstable modes develop. Under certain assumptions (such as Gaussian distribution of low-rank perturbations), there are exact links between outliers and nontrivial fixed points [5]. In general, however, eigenvalues that are outside the bulk, but not larger than 1 can also play a role in the dynamics. To capture this possibility, we devised the following measure for Figure 2c (top), $\sum_{i=1}^{N} w(\lambda_i)\lambda_i$ where

$$w(\lambda) = \begin{cases} 0 & |\lambda| <= 0.3 \\ |\lambda| - 0.3 & 0.3 < |\lambda| <= 1.3 \\ 1 & \text{otherwise.} \end{cases}$$

The value of 0.3 is given by the spectrum of the bulk - the expected distribution of eigenvalues from Wigner's circle law.

## 3.6 Rank-2 analysis

Analyzing the phase space of a high-dimensional system can be challenging. While it is possible to gain some understanding through the analysis of input-driven trajectories, it is not straightforward to access the entire dynamical landscape. However, when we work with the rank-2 version, we can plot and analyze the entire landscape by projecting every state $h$ onto $n$, and then analyze the system in the 2D system spanned by the columns of $m$—a plane we refer to as the $\kappa$-plane—instead of looking at the original $N$ dimensional space.

In the $\kappa$-plane, the neural speed can be redefined as:

$$q(k_t) = \|\kappa_{t+1} - \kappa_t\|_2, \quad t \in \mathbb{N}, \quad \kappa_{t+1} = n^T \tanh\left(\kappa m + b_{rec}\right)$$

In Figures 3 and 5, we analyze the dynamics as follows: We determine the limits of the $\kappa$ plane reached by the dynamics by considering $\kappa_{\max} = n^T \cdot \text{sign}(n^T \mathbf{1})$. We then evaluate the $q$ function over the two-dimensional grid $[-\kappa_{\max}^1, \kappa_{\max}^1] \times [-\kappa_{\max}^2, \kappa_{\max}^2]$ at every point. This allows us to display the level curves and verify to which slow region the dynamics of each task reaches, as shown in Figure 3D. In Figure 3D, we used a $q$-value of 0.1 to locate the curves. In Figure 5B, we used the level curves $0.1, 0.25, 0.5$, and $1$.

## 3.7   Repeating task-to-outlier analysis with other setups

We ran the same experiment of studying the relationship between number of bits and number of unstable eigenvalues in two other configurations. The first is Vanilla RNN, now initialized with a rich output regime: $W_{out}, b_{out}$ drawn from $U(-\frac{1}{N}, \frac{1}{N})$ instead of $U(-\frac{1}{\sqrt{N}}, \frac{1}{\sqrt{N}})$. The second is GRU networks, trained with the original regime. We trained the same number of networks in each of these configurations.

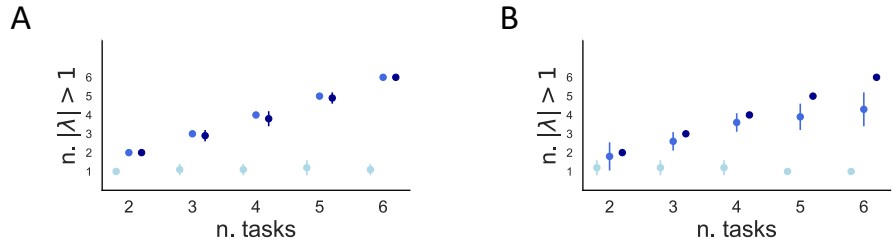

Figure 2: Repeating the analysis in fig.3B for both Vanilla networks trained on rich output regime (left) and GRU networks (right)

# 4   How input-output architecture shapes solutions

In the paper we addressed the fact that we chose an input-output architecture that imposes the minimal amount of constraints over the representation. Here, we describe three distinct configurations for input delivery and two for output delivery, showing how various combinations can lead to different results.

For input delivery, we consider three possibilities:

- A shared data channel + a transient context channel for each task. A transient context is a short pulse of magnitude 1 at the beginning of each trial.

- A shared data channel + a tonic context channel for each task. A tonic context is continuously delivered during the trial as a fixed value of magnitude 1.

- A separate data channel for each task. This setup was used in the main paper.

When a shared data channel is employed, the network must process inputs for all tasks uniformly. Especially in transient context scenarios, there is less available information during the trial to distinguish between the tasks.

For output delivery, we have two options:

1. A shared output channel

2. A separate output channel for each task. This setup was used in the main paper.

When the readout is shared across tasks, it imposes significant constraints on the representations. Specifically, all tasks must align with respect to their outputs.

We now hypothesize about how a network might jointly represent the attractors of two tasks.

- *Invariant* - across all tasks, inputs are aligned to exactly the same points.

- *Shared* - all tasks converge to the same attractor, but these are not identical to each other.

- *Equivariant* - some elements of the dynamics are shared while others are separate.

- *Separated* - each task is implemented in a distinct region of the state space.

Even though these options do not have to lie on a spectrum, they all differ in how much the manifolds are close to each other.

To demonstrate the role of architectural constraints in arriving at different solutions, we trained vanilla networks with each architecture on two tasks that correspond to the functions $\phi_1(v) = v^2$, $\phi_2(v) = (1-v)^2$. These functions produce opposite output sequences for the same uniform input interval $V$. Consequently, in state-space, the line attractors can align with respect to either input or output direction, but not both.

In Figure 3, we present the 2D PCA for each task, colored by the input value. It becomes evident that networks can align with respect to their inputs only when the output for each task is separate. Furthermore, it can be seen that when outputs are separate and context is transient, the network tends to lean towards a completely invariant representation. This is because a transient context is more similar to having no context at all, thus implementing all tasks simultaneously.

To avoid biasing the network towards completely separating or merging the tasks, we chose to focus on multiple inputs and multiple outputs in this work. Having multiple outputs and a tonic context signal is also feasible, but in this case, the attractors of the tasks reside on separate dynamical systems.

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

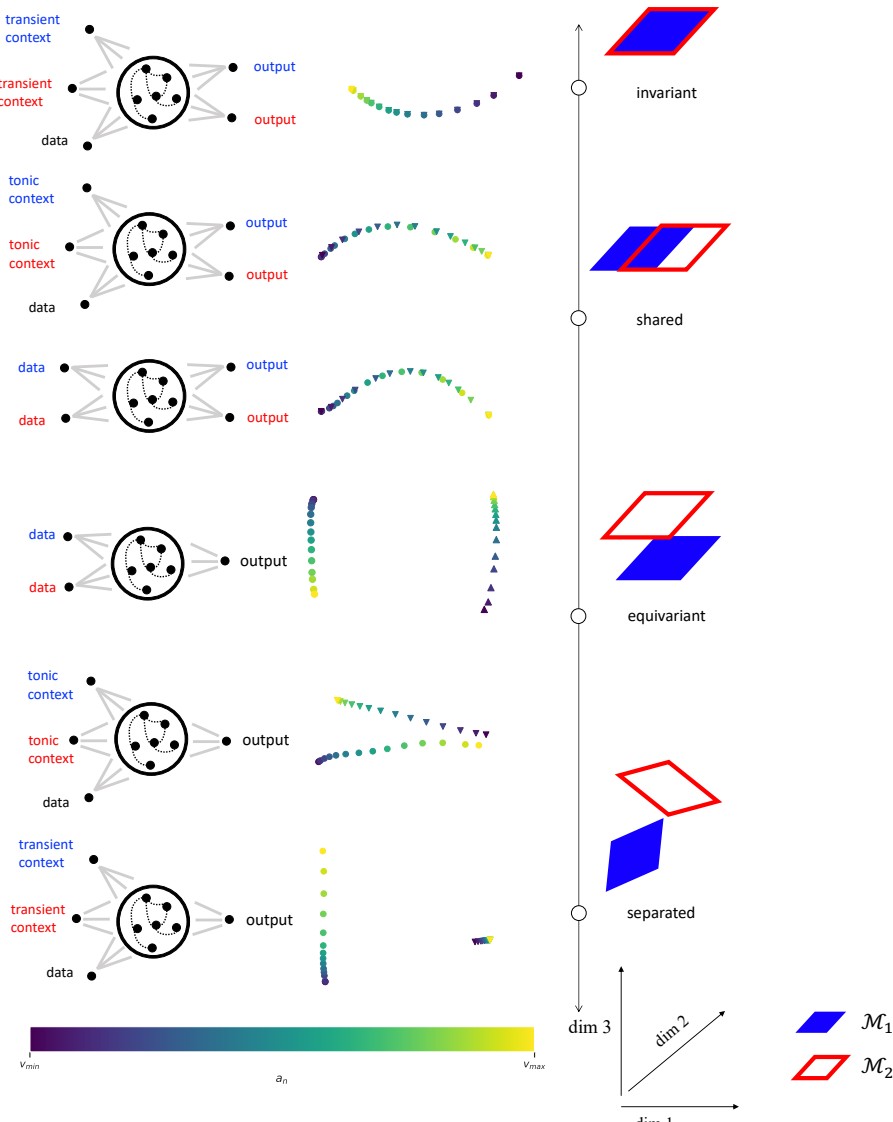

Figure 3: Qualitatively different joint representations obtained by different input-output architectures. Left: Six different architectures differentiated by either input delivery structure or output delivery structure. Middle: 2D-PCA for each architecture, trained on two complementary tasks. Right: Different potential joint representations of two neural manifolds corresponding to two tasks.