# OpenReview forum: "The Simplicity Bias in Multi-Task RNNs: Shared Attractors, Reuse of Dynamics, and Geometric Representation"
_NeurIPS.cc/2023/Conference — NeurIPS 2023 poster_

### Official Review · Reviewer_toNv · 2023-06-30

**Soundness:** 3 good
**Presentation:** 3 good
**Contribution:** 3 good
**Rating:** 6
**Confidence:** 3

**Summary:**

This paper studies how recurrent neural networks form shared attractors and reuse dynamics in the multitask setting. A simplicity bias is revealed, i.e., RNNs will not create new attractors unless necessary. The authors further investigate how task similarities (symmetry, gradients) can be translated to representation similarities. Finally, they discuss how their results are related to continual learning and modularity.

**Strengths:**

This paper poses an interesting scientific question on the formation of attractors in recurrent neural networks in the multi-task setting. The simplicity bias, to the best of knowledge, is novel and investigated in details. The experiments are nicely controlled, and evidence are clear and convincing. The paper is in general well-written. Figure 1 & 2 are pleasure to read (although Figure 3-5 need improvement, see weaknesses).

**Weaknesses:**

The scope of the paper is a bit limited: it only did experiments on toy synthetic datasets. Although this is probably fine for a scientific paper when one tries to understand something deeply, I feel that the science part hasn't been pushed to the limit:

(1) How do you expect your conclusion scale to more complicated datasets (e.g., contains 1000 tasks)?

(2) Can your analysis generalize to auto-regressive models (e.g., transformer-like language models)?

I feel that the impact of this paper on engineering needs more discussion.

(1) Is the simplicity bias a good or bad thing in practice?

(2) Once we understand the simplicity bias, what can we do to improve current RNNs?

**Questions:**

* Line 175, W_rec -> W_{rec}

* Figure 3, lack of axes make the plots hard to read. Figure 3D, I don't understand the difference between the 2nd and the 3rd subplot.

---

> ### Author Rebuttal · Authors · 2023-08-10
>
> Our work stems from neuroscience questions, but we thank the reviewer for causing us to speculate on possible engineering applications.
>
> Q1: How do you expect your conclusions to scale to more complicated datasets (e.g., contains 1000 tasks)?
>
> The concept of convergence to shared dynamical objects, and that new ones are formed only if the error does not decrease should apply there as well. We expect a more complex structure to emerge, perhaps hierarchical, governed by dynamical requirements.
>
>
> Q2: Can your analysis generalize to auto-regressive models (e.g., transformer-like language models)?
>
> The notion of dynamical objects is, to the best of our knowledge, not usually employed in transformer models. While intriguing, we feel that we cannot make educated speculations in this domain at this stage.
>
>
> Q3: Is the simplicity bias a good or bad thing in practice?
>
> The simplicity bias in neural networks is probably a double-edged sword. On one hand, it allows models to efficiently generalize across tasks that share underlying structures, reducing the computational overhead and potentially speeding up training. This can be especially beneficial when data is limited, or tasks are closely related. On the other hand, an overemphasis on simplicity could lead to suboptimal performance on distinct or more complex tasks, as the model might overly regularize or not capture unique nuances of specific tasks. Additionally, it could be less optimal to remove all information about the task in more realistic and complex settings, where the task requirements themselves change over time.
>
>
> One of our future directions is finding a set of tasks that afford two qualitatively different solutions, in order to compare their relative strengths and weaknesses in terms of generalization, performance, information-efficiency, robustness to noise, and flexibility.
>
> Q4: Once we understand the simplicity bias, what can we do to improve current RNNs?
>
> This depends on how we want them to generalize. Choosing the order in which tasks are trained can govern which dynamical objects are formed first, and serve as a scaffold for the following tasks. This, in turn, can shape representations, and the way in which networks will extrapolate.

---

> > ### Comment · Reviewer_toNv · 2023-08-10
> >
> > Thanks for the reply! I'll keep my original assessment.

---

### Official Review · Reviewer_yVHj · 2023-07-04

**Soundness:** 3 good
**Presentation:** 3 good
**Contribution:** 3 good
**Rating:** 6
**Confidence:** 4

**Summary:**

While the relationship between task requirements and neural dynamics in Recurrent Neural Networks (RNNs) has been studied for individual tasks, the dynamics of multiple tasks working together remain largely unexplored. The study introduces a systematic framework to examine multiple tasks in RNNs, minimizing interference from input and output correlations. The findings reveal that RNNs tend to share attractors and reuse dynamics, which is referred to as the "simplicity bias." It is observed that RNNs develop attractors sequentially during training, prioritizing the reuse of existing dynamics and opting for simple solutions whenever possible. Concrete examples demonstrate that new attractors primarily emerge due to task demands or architectural constraints, representing a balance between the simplicity bias and external factors. The geometry of joint representations along attractors is examined, and it is shown that task representations align based on their input strength, resulting in correlated projections. The research suggests potential applications, such as using the geometry of shared attractors to infer unknown tasks. Moreover, the simplicity bias implies that network modularity may not emerge spontaneously in RNNs without specific incentives, providing insights into the conditions necessary for network specialization.


**Strengths:**

These suggestions are intended to improve the overall clarity and accessibility of the research.

The paper provides the logical progression in the field from studying single tasks to exploring multiple tasks. This shift is considered more in line with the environmental conditions animals face, which often exhibit symmetries, regularities, and structures.

The study standardized the input and output structure across all tasks and used a consistent network architecture. This helped attribute differences in representations to intrinsic task complexities rather than variations in structure or design. Individual input and output channels were assigned for each task to prevent artificial convergence in representations. This ensured that any overlap or similarity in representations reflected shared computational requirements or reused dynamics across tasks.

The simplicity bias results is very interesting and I would like to encourage the authors to further develop and refine this intriguing finding.

The manuscript is overall clear and the relevant code was shared with the submission, permitting transparency of the results.


**Weaknesses:**

These suggestions are intended to improve the overall clarity and accessibility of the research.

The paper lacks a clear outline of its limitations, contributions, and literature reviews, and would benefit from providing more references to position the work within the context of prior research. It is also important to include information about the computational cost associated with the proposed approach.

The limitations outlined below highlight some of the weaknesses of the paper.

Regarding specific details:
* The legend for Figure 3B is incomplete and does not provide essential information. Additionally, the functioning of the axes in plot 3B is unclear.

* Figure 3C is not referenced or mentioned in the text, creating a discrepancy.

* The paper does not make any reference to the supplementary material, which should be included.

* Figure 5A is not properly referenced within the text, which may lead to confusion.

* The definition of 5D is missing, and its context should be clarified.

* Some typos and unclear sentences have been identified, which could benefit from clarification and proofreading.

* The readability of Figure 3D is poor, and it would be advantageous to increase its size to improve understanding.

To enhance the quality and readability of the paper, it is crucial to address these issues and make the necessary revisions.

**Questions:**

Dear authors, I would greatly appreciate it if you could respond to the following questions, as they would help improve my understanding of your work:

* Why are the representations learned in the parallel setting not displayed in the first figure?

* I am confused about how the two tasks are shown in the first setting, particularly with regards to "\0". My understanding was that in this setting, the network was trained on only one task. Could you please clarify?

* Line 277 suggests that the observed differences can be largely attributed to these architectures. I am unsure about the meaning of this statement. Could you elaborate further?

* Based on the findings, what conclusions can be drawn about neuroscience? How does this work contribute to our understanding of neural processes?

* Could you provide information about the color of the triangle in the figures?

Additionally, I find the results regarding the simplicity bias very interesting. I am curious to know if there could be a connection between this bias and the observed low rank bias in feedforward networks, where eigenvalues are learned sequentially when initialised with small random weights.

Lastly, I would like to know if you investigated the impact of different initializations in your experiments.

Addressing these concerns would greatly influence my evaluation.


**Limitations:**

To enhance the paper, it would be valuable to provide a clear outline of the limitations of the work. Some potential limitations to consider include:

*  Lack of biological validation: While the paper proposes mechanisms and patterns observed in RNNs, it may lack direct biological validation. It would be beneficial to explore experimental evidence or connect the findings to known neurophysiological phenomena to strengthen the biological plausibility of the conclusions.

* The other parameters of the network that could be looked at, I understand that for visualisation pursues and summarising  you look at subset of the information that could be looked at in this network.

* Scope of external factors: The paper acknowledges the influence of external factors on the emergence and dynamics of multiple tasks but may not extensively explore the full range of potential external factors. Further investigations into the interplay between these factors and the simplicity bias in RNNs could provide a more comprehensive understanding. For example how initialisation changes the observed dynamics.

Clearly outlining these limitations would contribute to the transparency and credibility of the paper, encouraging further research and discussion in the field.

---

> ### Author Rebuttal · Authors · 2023-08-10
>
> We deeply appreciate the reviewer's thorough examination of our work and the invaluable feedback provided. We acknowledge the areas of improvement, especially in references and technical accuracies, and commit to rectifying them diligently.
>
> Computational cost:
> The networks highlighted in this submission are compact (used N=100 hidden units), complemented by a modest-sized training dataset (512 trials x 200 steps). As a result, provided the training converges, the computational overhead remains minimal. For instance, a representative Vanilla network might require approximately 180 seconds for training — though this duration can fluctuate based on factors such as learning rate, batch size, and the set loss threshold — and utilizes around 16.2 MB of memory. In contrast, a comparable GRU network typically completes its training in about 110 seconds, maintaining a similar memory footprint
>
>
> Parallel set-up missing in fig 2:
> Overall, the more constrained the regime is, the harder it is to train. When all of the tasks are forced to be produced in parallel, and the task is not as “easy” as N-bit FF, Vanilla networks did not converge during training. They did converge on the orthogonal regime, which is more constrained than the don’t-care, but less than the parallel. This is why to exemplify the hybrids of the dynamical objects we focused on the first two regimes.
>
> The training regime of O:
> In this regime the network is trained on all of the tasks. The difference is, that when the network is tested on task 1, it is not “penalized” by the loss function on whatever it produced on channel 2. So each trial corresponds to only one task, but the training set consists of all of the tasks.
>
>
> Relevance to Neuroscience:
> One interesting idea that our work provides is that shared representation, or abstraction of the task in the neural dynamics, can be a result of laziness rather than sophistication. This is because in our experiments networks always developed a shared attractor before splitting it to multiple ones, and it could be that the nervous system uses the simplest representations when possible. Another relation is the order in which tasks are learned, that can affect the final representation achieved. The first task will determine the attractor that will serve as a scaffold for the following ones.
>
> Line 277:
> One major difference between our results and the results obtained in previous results are the input-output statistics and input architecture we used to train the network. We elaborate in the response to reviewer ZQ4N. Overall, since different papers have different approaches for how the information flows from input the output, such choices greatly affect the joint dynamics of the networks.
>
> Colors of shapes in figure 4:
> We edited the figure- it appears on the attached pdf.
>
>
> Connection between this bias and the observed low rank bias in feedforward networks:
> As mentioned in the discussion, there is a common mechanism in gradient descent dynamics - different structures evolve at different speeds. Because loss can be reduced in various ways, the fastest structure “wins” and the slower ones do not emerge. This is responsible for low-rank perturbations to connectivity in recurrent networks (Schuessler et al 2020), and is analogous to the neural race reduction (Saxe et al), and to our work.
>
> Different Initializations:
>
> Throughout our experiments, we trained networks using both the "Lazy" and "Rich" output regimes.
>
> For the Lazy regime, we initialized weights within the range U([-1/sqrt(N), 1/sqrt(N)]).
>
> For the Rich regime, weights were initialized within U([-1/N, 1/N]).
>
> In the tasks we explored, no significant differences were observed between the two initialization methods. Included in the attached PDF is a figure, showcasing results from networks trained under the "Rich" regime as a point of comparison to the "Lazy" approach.
>
> Definition and context of figure “5D”:
>
> We appreciate your attention to the detail. The reference was meant for Figure 5C-bottom. Our aim is to demonstrate that the geometry of tasks on the manifold is dictated by the relative 'spacing' or normalized derivative of the associated function. While earlier sections established that tasks with similar derivatives are proximate, this section elucidates why: tasks with analogous normalized derivatives align more closely in phase space due to consistent point-to-point proportions. Figure 5C-top visualizes this concept. In Figure 5C-bottom, we project each task to fit a regressor to the outputs of the other tasks. Tasks that are more similar in terms of their derivatives and spatial proximity exhibit similar structures, hence facilitating regression of their outputs.

---

> > ### Comment · Reviewer_yVHj · 2023-08-12
> >
> > Many thanks to the authors for your careful explanation and detailed rebuttal. I increased my score as a reflection of improvement in clarity and presentation as well as the general understanding of the contribution. I really appreciate the neuroscience focus of this paper.  In relation to neuroscience, it's worth noting that the inductive bias arising from the brain's connectomics/ network architecture could also potentially influence the attainable learnable attractors and representations.

---

### Official Review · Reviewer_ZQ4N · 2023-07-05

**Soundness:** 2 fair
**Presentation:** 1 poor
**Contribution:** 2 fair
**Rating:** 4
**Confidence:** 4

**Summary:**

This paper uses a multi-task RNN setup and tries to make sense of what computations get shared, and how they get shared.

**Strengths:**

A novel task set-up is used with gated, orthogonal, and parallel settings. The overall problem trying to be tackled is interesting.

**Weaknesses:**

1)	This paper is very confusingly presented. Section 2.2. is confusing. All the figures are extremely hard to parse and need much better annotation/labelling. I couldn’t make heads or tails what figure 3 was trying to say. Figure 4 suddenly has some triangles, pentagons, and diamonds. And they are different colours. Not explained why.

2)	A main claim is that an understanding of a ‘simplicity bias’ is provided, but it’s not clear what understanding is provided beyond Yang et al 2017, or Driscoll et al 2022. Overall, it’s not clear how this paper furthers our understanding beyond those papers.

3)	Only one network structure was analysed, while Yang, Driscoll, and others, show interesting phenomena under different architectural constraints.

4)	No reference is given to Saxe et al., 2022 which shows that shared representations are favoured as they are the ones which ‘see more data’ and so are learned fastest, thus explaining much of the phenomena from Yang, Driscoll, and your paper.

5)	(Reference [5] and [6] are the same paper.)


**Questions:**

See Weaknesses

**Limitations:**

Not clear that this clarifies anything over and above Yang or Driscoll. The paper is really hard to make sense of. Not because it is technically difficult, but because of its presentation. In particular, the figures are so under labelled that they may as well not be there.

---

> ### Author Rebuttal · Authors · 2023-08-10
>
> As written in the general response, we sincerely apologize for the lack of clarity. We hope the new figures 3 and 4 convey our first steps towards resolving this issue, and we will make this a high priority for the final manuscript.
>
> Relation to Yang & Driscoll. We indeed drew inspiration from these studies, but our approach extends and diverges from them in several respects.
>
> Input-output structure: The task set used by Yang et al (and Driscoll et al) was inspired by tasks used in neuroscience experiments. Both works cleverly utilized the resulting networks to ask questions about shared representations and dynamical objects - that partially overlap with our work.
> The neuroscience tasks exhibit correlations between input-output statistics and cognitive demands. For instance:
> Input statistics: Working-memory tasks typically involve two stimuli separated by a delay, while integration tasks use continuous stimuli.
> Output statistics: Decision-making tasks generally involve a binary outcome, whereas the classic GO task necessitates input stimulus replication.
>
>
> Input-output architecture:
> Yang's study was potentially constrained by using a shared output channel. Our methodology avoided this pitfall by ensuring outputs were distinct across tasks, so that the representations are not forced to be aligned w.r.t. the output of the network. Otherwise, tasks are forced to be shared or separated depending on whether they require similar or different outputs. In section 4 of the supplementary material, we demonstrate a spectrum of input-output architectures and how each of them constraints the joint representation in various ways.
>
>
> First-Principle Approach with Basic Dynamical Objects:
> A foundational aspect of our work is the first-principle approach to understand the dynamics of RNNs. Our rationale was that in order to understand dynamical reuse of objects in complex tasks, we should first focus on primary dynamical objects such as fixed points, limit cycles, and line and plane attractors. It is unclear a priori whether two topologically distinct dynamical objects will merge in a multi-task set up. We showed that networks tend to merge these primary dynamical objects into singular representations, suggesting that this simplicity bias might be a general principle extending across various object types.
>
> By supplying three different task regimes, where any solution of the last two is also a solution to the first, we show that tasks share objects when possible, but when constrained develop separate solutions. Hence, representation sharing is a bias but not definite, and we show that task regime is one way to force separation. Another way to sculpt the joint representation is via the input-output architecture, as discussed in Supp. 4.
>
> Nuanced Insights into Learning Dynamics:
> We show that the number of dynamical objects is correlated with the number of unstable eigenvalues of the recurrent matrix. We show how learning, in our regime, starts with no unstable modes and the origin as the only attractor. Both the number of dynamical objects and the number of unstable values commonly increase with training, and that networks start with a shared dynamics solution before developing additional unstable directions.
> So, we provide an argument for the preference for shared representation.
>
>
> Dive into Representational Geometry:
> We ask another set of questions about the geometry of the shared representations, and check which task similarity correlates with similarity in representation. To do so, we focus on a set of tasks that all require line attractor, and only differ in the mapping between input and output.
>
>
> Reference to Saxe et al. 2022:
> This reference is indeed pertinent, and it was an oversight on our part not to include it in line 281. Our findings align with the 'race' concept presented in that study, consistent with the other works referenced in that section. However, our experimental setup differs: we employ recurrent networks, leading to a race in the formation of dynamical objects, rather than connections between modules as in a feedforward network. Schuessler et al.'s research might be more analogous, given that their eigenvalue outliers emerge sequentially at varying rates. We do believe in the generality of the race analogy, and therefore cited works from several different domains in that paragraph.

---

> > ### Comment · Reviewer_ZQ4N · 2023-08-13
> > **Many thanks for your response**
> >
> > Many thanks for your response. I will raise my score based on the your statements that you will hugely increase the clarity of the paper. I am still not convinced how much this paper offers over Yang/Driscoll, but now see some places - many thanks.

---

### Official Review · Reviewer_6PJE · 2023-07-08

**Soundness:** 4 excellent
**Presentation:** 3 good
**Contribution:** 3 good
**Rating:** 7
**Confidence:** 3

**Summary:**

The paper proposes "simplicity bias" in RNNs when learning multiple tasks simultaneously. In particular, the paper focuses on investigating the formation of attractors in the dynamic system of RNN when tasks with variant difficulties are handled jointly. The RNN develop attractors sequentially, and simple attractors will emerge first, and new attractors may organize with task demands or architecture constrains. The authors conducted extensive, carefully-designed numerical experiments to demonstrate their ideas. The findings shed light on understanding the dynamics of artificial RNNs and also cortical functions in the brain.

[post-rebuttal]
I appreciate the authors' reply. I recommend an acceptance.

**Strengths:**

- The experiments are well-designed
- Analysis is conducted from various perspective
- The results are visualized and presented in an intriguing way
- Too my knowledge, the insights about simplicity bias in RNNs is novel (while there are a few works about other kinds of models, but the not from a dynamical system perspective).

**Weaknesses:**

Overall, I think the paper did a good job on demonstrate the core concept of simplicity bias in RNNs. If there is something missing, I would say that it is unclear to what extend the conclusions apply. The tasks being tested are simple enough to understand, which contributes to more intuitive understanding for readers. However, the real-world tasks are usually much more complicated and it is hard to define or compare the diffulties of tasks.  One particular design about model architecture is that the input and output channels for individual tasks are distinguished, which may not be the case for the brain or many ANN models. In sum, there remians much to do to further validate and understand the conclusions made in this paper.

**Questions:**

- The abstract reads a bit too long and redundant, might it would be more concise to limit the absract to 200 words.
- Reference 5 repeats with 6
- Although I think it is reasonable to first consider the basic Elman-type RNN (eq. 1), I am also curious about whether other kinds of RNN such as LSTM and GRU features with similar simplicity bias, or there is something different.
- Some figures (e.g., Fig.2) is too cotmpact which makes it hard to read when printed on paper.

**Limitations:**

This work is foundamental and rather like a proof of concept. Future work should try to clarify how much the conclusions can extend to broader model architectures and task properties.

---

> ### Author Rebuttal · Authors · 2023-08-10
>
> We thank the reviewer for the positive evaluation of our work.
> Specific comments have been addressed in the general response to all reviewers, encompassing areas such as study limitations, GRU and LSTM discussions, and clarity aspects (e.g., Figure 2). We will refine the abstract to enhance its precision and conciseness. Thank you for the valuable suggestion.

---

> > ### Comment · Reviewer_6PJE · 2023-08-11
> >
> > Thanks for the reply! I keep the original recommendation.

---

### Author Rebuttal · Authors · 2023-08-09

We thank the reviewers for their constructive comments. Responses to common themes are here, and detailed individual responses are below.
Clarity: We sincerely apologize for the lack of clarity noted. We have already started clarifying the figures (examples in the 1-page PDF), and will continue doing so along with the text for the final version.

Technical Corrections:
Thanks for pointing out the technical issues in the text. We fixed them in the manuscript.

PDF:
We attach the newer versions of figures 3 and 4 from the paper, along with detailed captions.

Studying other architectures (GRU/LSTM):

Initially, we studied both GRUs (Gated Recurrent Units) and LSTMs (Long Short-Term Memory networks) in our experiments. These architectures, renowned for their enhanced memory capabilities, have shown rapid and efficient learning in our tasks. However, as we delved deeper into understanding the intrinsic properties of neural dynamics, certain features of these advanced architectures emerged as potential obstructions.

The strength of GRUs and LSTMs lies in their intricate internal gating mechanisms. GRUs, for instance, comprise three distinct gates, with the update gate being of particular interest. This gate allows the network to preserve its memory from previous time steps by choosing to propagate forward the input from past states, effectively resisting an immediate update based on new information. This "immunity" to immediate synaptic updates, although computationally powerful, deviates from our understanding of biological neural dynamics. During our experiments, we noticed that many networks leaned on a very small number of neurons as persistent task-bits, allowing them to maintain specific states indefinitely.

While this property is valuable for certain computational tasks, it poses questions about biological plausibility. Real biological neurons, as we understand them, do not exhibit such prolonged immunity to updates. Therefore, using architectures that permit such behaviors could lead us away from capturing foundational properties of biologically plausible dynamics.

In contrast, the Vanilla RNN, while more straightforward, offers a clearer lens through which we can study these foundational properties. It provides a more direct and interpretable mapping between input, internal dynamics, and output, making it better suited for our study's objectives.

For completeness, we attach a version of fig 3B that was generated with GRU networks, showing qualitatively similar behavior to vanilla networks.

Limitations of our study:

We appreciate the call to delineate the constraints of our research. Indeed, there are several factors that merit attention. We will also make sure these considerations are integrated into the discussion.


Biological Relevance: Even though our findings could hold implications for biological neural networks, drawing direct parallels remains a challenge. The architecture and function of biological networks are vastly more intricate, involving unclear connectivity, immensely differing scales of dimensionality, and overlapping neural processes, notably synaptic plasticity. Our study did not venture into multi-layer configurations, which might provide a more apt comparison to layered structures within the brain.
Network Parameters: The behavior of neural networks is inherently influenced by numerous free parameters. While we addressed several of these parameters, there remain others not explored in our study. For instance, while network size appeared to have a negligible impact on our results, the introduction of other activation functions or regularization techniques compromised the efficacy of training Vanilla networks.
Task Setup: Our approach to multi-task learning was crafted with systematic rigor, yet it represents just one among myriad potential configurations. Achieving a holistic understanding of multi-task representations may necessitate a combination of analytical methodologies, more exhaustive task setups, or a diversified array of such configurations.
Scalability: Our investigations were conducted on a specific scale with respect to the number of tasks and network size. It remains uncertain how our findings would extrapolate to more expansive datasets comprising hundreds or thousands of tasks. Additionally, while our networks effectively handled the tasks presented, they might exhibit different dynamical and representational behaviors when scaled to accommodate more complex task sets or larger architectures. The relationship between scale and the observed simplicity bias is an area ripe for further exploration

---

### Decision · Program_Chairs · 2023-09-21

**Decision:**

Accept (poster)

**Comment:**

The paper investigates simplicity biases in RNNs, such that they re-use dynamical elements like attractors to perform new tasks.

Strengths:

-Well designed experiments

-Interesting analyses from a variety of angles that yield greater insight into the mechanisms and nature of re-use

-Important question and problem setting

Weaknesses:

-Should better differentiate findings from related work on RNNs and contrast with related phenomena in feed forward networks. In particular, the idea of studying multitasking in RNNs is not new (Yang et al., Driscoll), nor is the idea that dynamical features can be reused; the novelty of this paper is based on the specifics of its setting and analyses

-Clarity could be improved, with greater discussion of limitations, contributions, and literature review

-Experiments are with small datasets and it is unclear to what extent the insights will scale (as with much work of this type at present)